# Application of Seismic Fragility of Buried Piping Systems with Bellows Expansion Joints

**Joon-Il Ryu [1], Bub-Gyu Jeon [2], Ho-Young Son [1],\* and Bu-Seog Ju [1],\***

[1]  Department of Civil Engineering, Kyung Hee University, Yongin-si 17104, Republic of Korea
[2]  Seismic Research and Test Center, Pusan National University, Yangsan-si 50612, Republic of Korea
\*  Correspondence: shyoung0623@khu.ac.kr (H.-Y.S.); bju2@khu.ac.kr (B.-S.J.)

**Abstract:** Bellows expansion joints are known to have a large displacement capacity and can thus be potentially used to improve the seismic performance of buried piping systems. However, there are no guidelines on the installation of bellows expansion joints for the seismic performance improvement of buried piping systems. Furthermore, there are very few studies on the seismic performance of buried piping systems with bellows expansion joints. In this study, therefore, we performed seismic fragility analysis according to the installation conditions to obtain basic data for the installation guidelines of bellows expansion joints. Therefore, in this study, an experimental test was performed on bellows expansion joints considering the characteristics of earthquake loading conditions, and a 3D finite element (FE) model using the ABAQUS platform was developed and validated based on the experimental results. This model was verified by comparing the force-displacement relationship and energy dissipation. Leakage occurred at a displacement of 113.6 mm in the experiment, and the FE analysis result was also applied up to the same displacement. In the case of energy dissipation, an error between the FE model and experimental result was determined not to be significant. However, the appearance of such physical performance errors is due to the manufacturing errors resulting from the bellows forming process and the variability of material properties. Finally, seismic fragility analysis of buried pipeline systems with bellows expansion joints was performed. In addition, the following cases were used for analysis according to whether bellows were applied or not: (1) without a bellows expansion joints; (2) with a single bellows expansion joint; and (3) with two bellows expansion joints. In conclusion, it was found that the seismic performance of the buried pipeline system was improved when bellows were applied. However, the effect of the seismic fragility curve according to the increase in the number of bellows was insignificant.

**Keywords:** buried pipeline system; bellows expansion joint; cyclic loading test; FE model; seismic fragility analysis

## 1. Introduction

The buried pipeline system is used to provide essential energy sources required for industrial facilities, and it is generally installed with a large, long-scaled straight piping system and connected with various types of joint conditions, such as threaded, grooved, and bellows expansion joints. Damage to a buried pipeline systems caused by external events such as strong earthquakes may cause disruption of the operations of critical industrial facilities leading to secondary damage such as explosions and floods. Therefore, the damage induced by strong ground motions on buried pipeline systems must be mitigated for continuous energy supply and stable industrial facilities operations. In order to minimize the damage to the piping system, sufficient seismic performance in particular must be secured as the buried pipeline system is likely to be frequently exposed to natural disasters. Earthquake-induced ground subsidence and liquefaction damage buried pipeline systems, acting as a large relative displacement. Such phenomena revealed that damage to the buried pipeline system caused by large relative displacement

generally occurs in connections such as joints and fittings [1]. Therefore, many studies have recently addressed the evaluation of the seismic performance of the joints and fittings (e.g., elbow and tee) of pipeline systems. First, a quantitative ultimate state evaluation was conducted for the steel elbow piping system applied welding conditions in consideration of large relative displacement [2–4]. Next, an experiment was conducted to evaluate the low-cycle fatigue behavior of steel pipe tees [5]. A study on seismic behavior analysis and development of the FE model for a non-welded pipeline using the stainless pressure joint was also conducted [6]. An experimental study of PVC pipes was conducted to develop seismic performance and subsidence resistance corresponding to material performance [7]. Studies on seismic performance evaluation considering soil properties, fault mechanisms, and buried conditions were also conducted as gas pipelines are generally installed and buried in the ground. An analytical-based seismic fragility assessment of the buried pipeline system was performed, and the effects of buried depth, soil conditions, and boundary condition parameters were analyzed [8,9]. A seismic fragility assessment was performed for a pipeline system installed crossing different ground properties based on peak ground velocity (PGV) [10]. Fragility evaluation of the pipeline system in the case of permanent ground deformation was performed considering ground and burial conditions via machine learning [11]. Furthermore, a numerical analysis of the buried pipeline system was performed to calculate seismic fragility and economic loss predictions [12].

The bellows-type expansion joint is a structure composed of several connecting thin convolutions. It has a large displacement capacity as its geometrical characteristics allow some axial and angular deformation [13]. An experimental test and FE analysis of the buried pipeline system with bellows were performed to evaluate the effect of reducing damage caused by fault rupture [13,14]. A study analyzing the bellows' load capacity and energy dissipation was conducted using the FE method [15]. This study observed the load-displacement relation according to the loading condition change and the convolution number. A numerical analysis was performed to evaluate the mechanical behavior caused by fault rupture of the hinged bellows type [16,17]. This study modeled the pipeline system using a beam element and replaced soil properties with spring stiffness. The numerical analyses of existing studies suggest that bellows expansion joints can improve seismic performance. However, there is little research on the seismic performance evaluation of the bellows expansion joints considering repetitive and large relative displacement, a characteristic of seismic load. Performance evaluation through experimental tests considering seismic load characteristics is necessary. According to previous studies, a seismic wave due to soil properties and deformation can affect the buried pipeline system [18–20]. Additionally, there are only a few studies on the probabilistic safety evaluation of buried pipeline systems with applied bellows expansion joints. Current deterministic safety assessments cannot consider material, input motion, and soil condition uncertainties. An earthquake involves uncertainties such as fault mechanism, frequency, and peak ground acceleration (PGA). Therefore, a probabilistic seismic safety evaluation considering the soil-structure interaction of the buried pipeline system is needed.

The buried pipeline experimental test considering soil conditions involves high costs and limited space. In particular, numerical research using the FE model is effective for parameter analysis and probabilistic safety evaluation that require a large amount of data. Development and verification of an FE model based on experimental test data are necessary for numerical analysis in a high fidelity simulation. Therefore, the present study performed a multi-step increasing amplitude cyclic loading test by applying a loading protocol in which the amplitude gradually increases, taking into account the bellows expansion joint's seismic load characteristics. The FE model was developed and verified based on the experimental test data. Probabilistic seismic fragility analysis was performed using the buried pipeline system FE model to which a verified bellows expansion joint was applied. Soil-structure interaction was considered and was compared with the straight pipeline system to observe the change in seismic performance according to the application of the bellows expansion joint.

## 2. Cyclic Loading Test of a Bellows Expansion Joint

### 2.1. Description of a Bellows Expansion Joint

A cyclic loading test was performed to evaluate the seismic performance of the bellows expansion joint. Bellows-forming methods include mechanical and hydroforming. They are classified into U type, S type, and Ω type, depending on the circuit type. The target bellows were two-ply, hydroformed U-type bellows. Figure 1 shows a schematic of the target bellows. w, q, and t denote the height of the convolution, interval of the convolution, and thickness of the bellows, respectively. There are six convolutions, and the data are summarized in Table 1. Figure 2 shows an illustration of the bellows expansion joint. The bellows had a 110 mm overall length and had SS275 carbon steel pipe welds on both ends. The 19 mm thick flanges on the ends of the pipe were used to connect it to the jig and actuator. The pipe was 4 mm thick. For bolted connections, the flange had four holes that were each 19 mm in diameter. Gas supply pipes in residential areas are typically 100 mm or less. Therefore, the study was conducted using 80A (D = 89.1 mm) pipes.

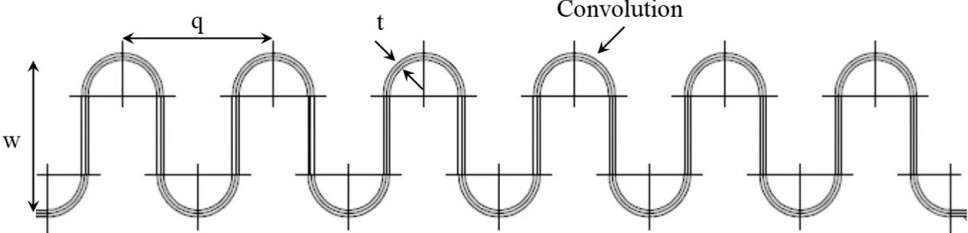

**Figure 1.** Description of 2-ply bellows.

**Table 1.** Bellows properties.

| Description | Value |
|---|---|
| Height of Convolution, w | 15 mm |
| Pitch, q | 15 mm |
| Thickness, t | 0.6 mm |
| Number of Convolutions | 6 |

Unit: mm

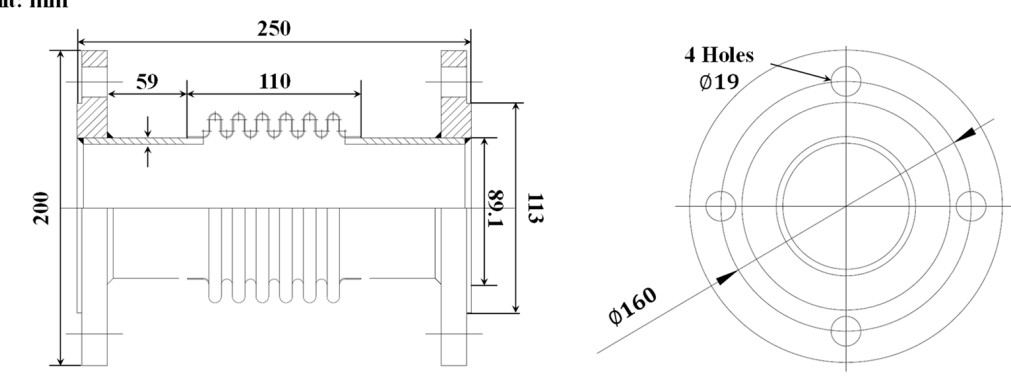

**Figure 2.** Drawing of bellows expansion joint.

### 2.2. Cyclic Loading Test

The buried pipeline system is susceptible to damage from ground motion, which could result in the loss of energy sources. Large relative displacements and repeated loads that occur during earthquakes mainly damage the joints in pipeline systems. Although the large displacement capacity of the bellows expansion joint is well known, there is limited experimental research supporting this. Therefore, this study performed a cyclic loading test of the bellows expansion joint.

ANSI/FE Approvals 1950 [21] presents the design and performance requirements for seismic sway bracing components and assemblies. The cyclic test on components gradually increases the force after the initial force is repeated 15 times. The test is performed while maintaining a frequency below 0.1 Hz and continues until the test subject is destroyed. KB S 1528 [22] refers to ANSI/FE Approvals 1950 [21] and proposes a cyclic test method for pipe connections, including fittings and joints. It is recommended to determine the amplitude using the displacement control method because the main source of pipe connection damage from earthquakes is repetitive and has large relative displacement. The test must be terminated when a component suffers damage or leaks. Pipe connections are damaged by the structure's relative displacement brought on by an earthquake. Therefore, this study used the KB S 1528 [22] cyclic test method to conduct the test. The loading amplitude (Δl) repeats the initial displacement (X) 15 times the number of cycles (N), as in Equation (1). Then, the amplitude gradually increases until leakage occurs, as shown in Equation (2). Figure 3 shows the loading history created using Equations (1) and (2).

$$\Delta l = X, \quad \text{for N} = 15 \text{ cycles,} \tag{1}$$

$$\Delta l = X \times \left(\frac{15}{14}\right)^{\frac{(N-15)}{2}}, \quad \text{for N} > 15 \text{ cycles,} \tag{2}$$

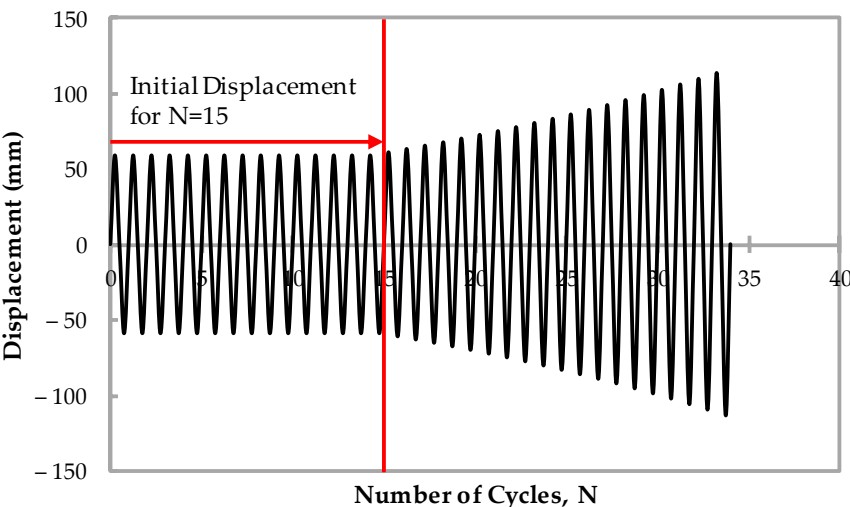

**Figure 3.** Loading protocol for the cyclic test.

Figure 4 illustrates how the test was prepared. Figure 4a shows the test setup. An LM guide was installed to constrain the degrees of freedom beyond the loading direction. A pin connection between the LM guide and the specimen and the bending deformation on the bellows was implemented to achieve this goal. The test was designed to end when the bellows were damaged and began to leak. For this purpose, the specimen was filled with water. It was pressurized using an air booster, and an internal pressure of 0.4 MPa was maintained during the test [23]. The universal testing machine (UTM) was used to load the vertical cyclic displacement. Load and displacement were measured using a load cell and LVDT.

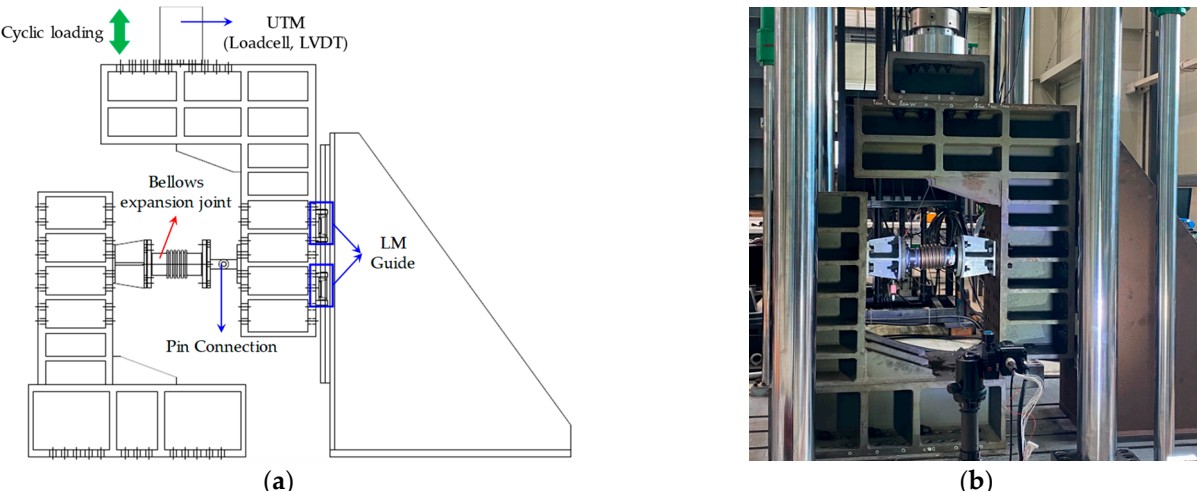

**Figure 4.** Test setup: (**a**) test configuration; (**b**) test photo.

## 3. FE Model and Validation of a Bellows Expansion Joint

Only a few of the longer pipelines have undergone experimental tests to evaluate their seismic performance. This is because of the limitations of equipment, such as actuators and shaking tables, which can simulate and test a large-scale buried pipeline system. Additionally, the cost and time required to conduct many numerous experimental tests are high for probabilistic safety evaluation and parameter analysis. Furthermore, parameter analysis requires considerable data. In contrast, numerical and analytical studies are efficient and free of various limitations. However, reliable research is possible only when the numerical model is modeled and verified using design data and experimental tests. Additionally, a material model for simulating the nonlinear behavior of a material must be determined exclusively through tests. Therefore, this section describes and verifies the tensile coupon test of the materials used to construct the bellows and the FE model of the bellows expansion joint.

### 3.1. Tensile Coupon Test for STS 316L

To develop a reliable finite element model, the material model was determined by performing a tensile test of STS 316L, which is used for manufacturing bellows. The specimen's width (w) and thickness (t) were measured at the locations shown in Figure 5. Area calculations were used to determine the average across all locations, which are listed in Table 2. The test was conducted using three constructed specimens. A strain gauge was attached to the specimen's center, as shown in Figure 6. An extensometer was additionally installed to collect reliable data. True stress ($\sigma_{true}$) and true strain ($\varepsilon_{true}$), considering the change in the cross-sectional area, must be applied to the material properties input into the ABAQUS platform [24]. Therefore, the nominal stress ($\sigma_{nom}$) and nominal strain ($\varepsilon_{nom}$) measured during the test are converted into true stress and true strain using Equations (3) and (4).

$$\sigma_{true} = \sigma_{nom}(1 + \varepsilon_{nom}), \tag{3}$$

$$\varepsilon_{true} = \ln(1 + \varepsilon_{nom}) \tag{4}$$

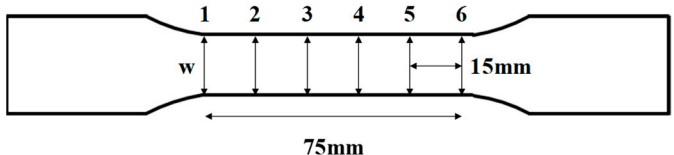

**Figure 5.** Locations at which the specimen width (w) and thickness (t) were measured.

**Table 2.** Dimension average for each specimen.

| Specimen | Thickness (t, mm) | Width (w, mm) | Area (A, mm) |
|---|---|---|---|
| S1 | 1.99 | 23.94 | 47.68 |
| S2 | 1.98 | 23.89 | 47.39 |
| S3 | 1.98 | 23.90 | 47.24 |
| S4 | 2.00 | 23.92 | 47.83 |

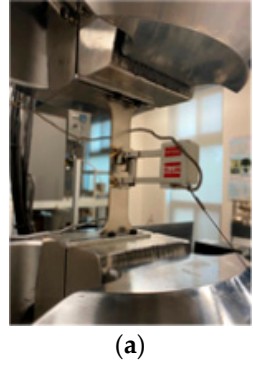
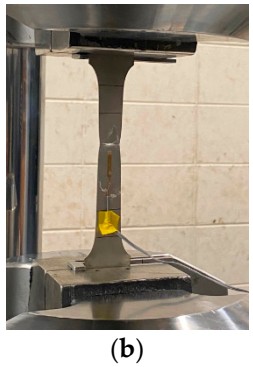

(**a**)      (**b**)

**Figure 6.** Tensile test of STS 316L: (**a**) test setup and extensometer; (**b**) result.

The true stress and true strain of each specimen converted using Equations (3) and (4) are shown in Figure 7. The modulus of elasticity was estimated to be 188,000 MPa using the least squares method in the true stress–true strain relationship. A kinematic hardening model was assumed for the material's hardening. Since the kinematic hardening model is assumed to be a bi-linear type on the ABAQUS platform [24], the yield and ultimate stress were determined in a such way that the error from the true stress–true strain relation obtained in the experiment would be less than 2%. The yield and ultimate stress were 360 MPa and 1050 MPa, respectively.

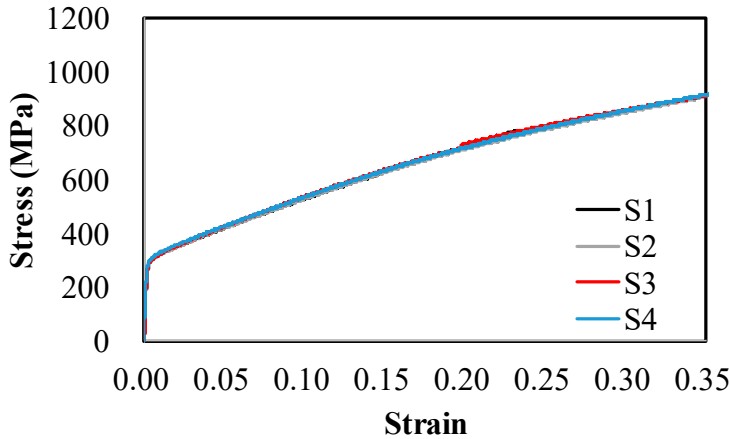

**Figure 7.** Stress–strain relationship for each specimen.

### 3.2. FE Model for Bellows Expansion Joint

An FE model was developed using the ABAQUS platform [25] based on the cyclic loading test data of the bellows expansion joint. The bellows expansion joint FE model consists of bellows, pipe, flange, bolt, and jig. The bellows used a reduced 4-node shell element (S4R), whereas the jig used a 2-node beam element (B31). A reduced 8-node solid element (C3D8R) was used for the pipes, flanges, and bolts. The element types and numbers used in each component are summarized in Table 3. The bellows' mesh size, where nonlinear deformation is expected to be concentrated, was set to 1.5 mm. The mesh size of pipes, flanges, and bolts was set at a relatively arbitrary 5 mm. Figure 8a shows the mesh

shape of the FE model, whereas Figure 8b shows the loads and boundary conditions applied to the FE model. An internal pressure of 0.4 MPa was applied considering the weight of the bellows expansion joint by applying gravity in the +x direction. A cyclic load was applied to the end of the jig in the x direction while maintaining internal pressure and gravity. The ends of the flange and jig, respectively, were subjected to fixed and pin conditions. The coupling option was used between the flange and the jig, and the tie option was applied to simulate the welding connection between the bellows and the pipe. A contact condition and a friction coefficient of 0.2 were both used to consider the interaction between the flange and the bolt. The self-contact condition was applied while considering frictionless contact because each convolution of the bellows could collide owing to repeated load.

**Table 3.** Element properties of the FE model for the bellows expansion joint.

| Component | Type | Number of Elements | Number of Nodes |
| --- | --- | --- | --- |
| Bellows | S4R | 29,200 | 29,400 |
| Pipe and flange | C3D8R | 16,827 | 23,018 |
| Bolt | C3D8R | 3584 | 5032 |
| Jig | B31 | 10 | 11 |

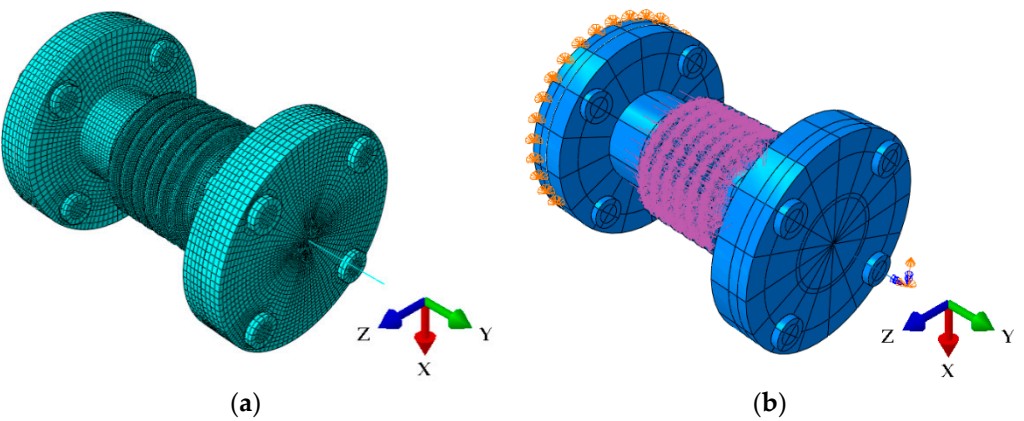

(**a**)　　　　　　　　　　　　　　　　　　　　　(**b**)

**Figure 8.** Finite element model for bellows expansion joint: (**a**) mesh shape and (**b**) load and boundary conditions.

### 3.3. Validation of the FE Model

This section compares the results of the force-displacement relation, energy dissipation, and FE analysis to validate the developed model. Figure 9 compares the force-displacement relation of the experimental test and FE analysis. In the experiment, the force in the plus direction showed a tendency of decreasing gradually in the unloading process while the initial displacement was being applied. The maximum force of the first cycle was measured at 1.91 kN, but that of the 15th cycle was measured at 1.10 kN. Although the maximum force decreased by up to approximately 42% at the initial displacement, no decrease in the force was observed in the FE analysis result. During the hydroforming process of the bellows, wall-thinning of convolution may occur, and the material may experience plastic deformation [26]. Variations that may occur during the fabrication process can affect the mechanical behavior of bellows. Therefore, the maximum force may decrease as the number of cycles increases. However, in the case of FE analysis, it seems that no decrease in force was observed because the thickness and material properties of the bellows were applied based on the design conditions. After the 15th cycle, the displacement increased gradually, reaching 113.6 mm, and the experiment ended because of leakage in the convolution of the bellows during the unloading process. In the FE analysis, too, the calculation was terminated because of the excessive plastic deformation during the unloading process after reaching 113.6 mm. A difference in the response occurs between the experiment and analysis because of the uncertainties that may occur in the forming process. Furthermore,

it seems that a lower force is generated in the experiment than in the FE analysis because of the Bauschinger effect and the spring-back phenomenon. Differences in compressive and tensile stresses occur during the loading and unloading processes, and this effect is called the Bauschinger effect [27]. Additionally, the energy dissipation occurring in each cycle was calculated, as shown in Figure 10. In most cycles, the calculated energy dissipation in the FE analysis result was larger, with a difference of up to approximately 10%. As the cycle increased, the error decreased gradually, and the energy dissipation of the last cycle was almost the same. In sum, the developed FE model was validated based on the experimental data, and it seems that it can sufficiently simulate the dynamic behavior of the bellows.

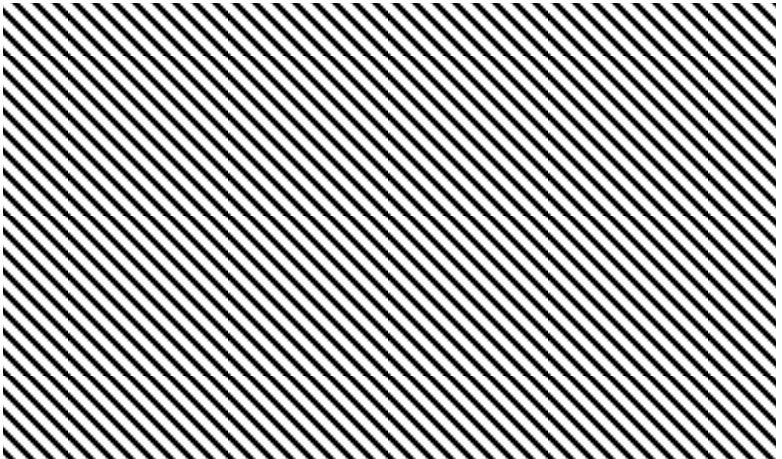

**Figure 9.** Force-displacement relation obtained from experimental tests and FE analysis.

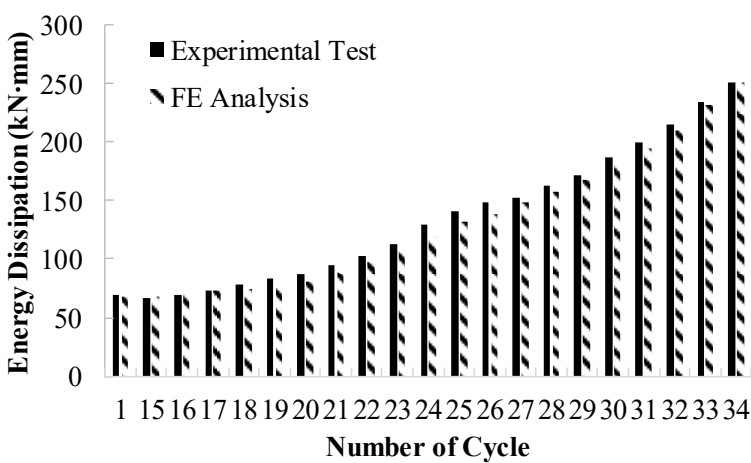

**Figure 10.** Energy dissipation for each cycle obtained from experimental test and FE analysis.

## 4. Development of Soil-Bellows Piping System Interaction FE Model

The vibration spread can affect the piping system's mechanical behavior owing to the ground characteristics. Therefore, studies have been conducted to evaluate seismic performance considering soil-piping system interaction. Soil parameters and the soil-piping system interactions are often replaced with linear or nonlinear spring elements to reduce computation costs and simplify the model. However, applying a spring element to a rough mesh size had the disadvantage of causing local buckling or making it impossible to predict the geometric nonlinearity of the pipe wall [28]. Therefore, this study developed a Soil-Bellows piping system interaction FE model considering the solid element and contact conditions.

In previous studies, performance evaluation was performed on buried piping systems with bellows expansion joints under repeated compressive and shear loads [12,13]. These studies targeted a short-length buried piping system and evaluated the performance

based on the repeated loads of the piping system with and without bellows. However, there are no specific criteria or recommendations for the installation conditions (such as position or quantity) of bellows expansion joints in a long-connected buried piping system. Therefore, we conducted this study as a basic study for providing installation guidelines of bellows expansion joints and analyzed the seismic performance considering the following installation conditions of the bellows: (1) without a bellows expansion joints; (2) with a single bellows expansion joint; and (3) with two bellows expansion joints. Figure 11 shows the FE model of the piping system according to the installation conditions. Its total length is 300 m. All pipes excluding bellows were assumed to be SS275 carbon steel pipes. According to Kim et al. [29], the elastic modulus, the yield stress, and the tensile strength of SS275 carbon steel were estimated to be 216,000 MPa, 300.9 MPa, and 445.46 MPa, respectively. For the linear pipe, a reduced S4R with a mesh size of 1.5 mm in the hoop direction and 500 mm in the longitudinal direction was used. The thickness of the linear pipe was assumed to be 4 mm, which is the typical thickness for an 80A carbon steel pipe [30].

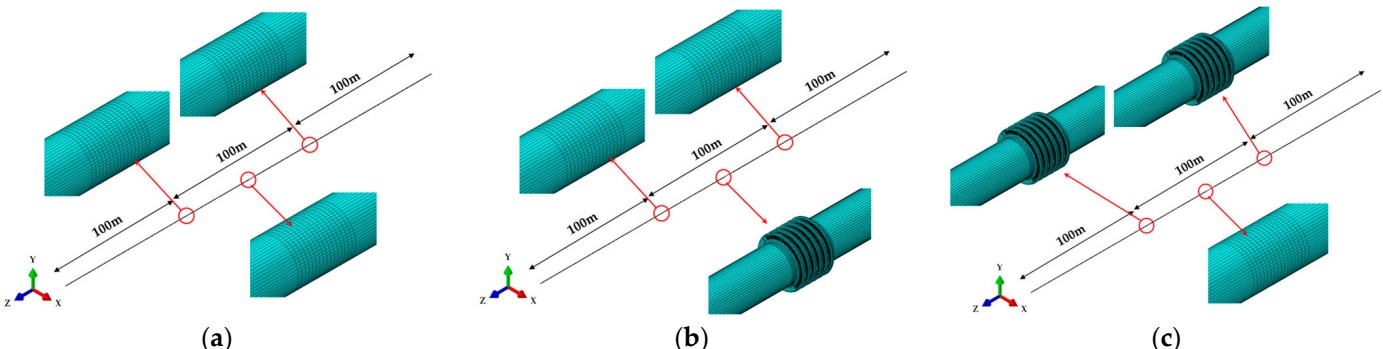

**Figure 11.** FE model of the pipeline system and joint location: (**a**) no bellows; (**b**) single bellows at the center; and (**c**) two bellows at both sides.

The soil's total length was 300 m and its cross-section was 600 mm × 600 mm. Figure 12 shows the isometric and section views of the soil FE model. Most of the soil used a reduced C3D8R, while some regions used a reduced 6-node solid element (C3D6R) owing to the convolution form of the bellows. The mesh size of the pipe's periphery was applied densely at 42 mm, and the other mesh sizes were applied roughly. The burial depth was assumed to be 1300 mm, considering the general burial depth of gas pipelines [23]. The coefficient of friction between the soil and the pipeline may vary based on the longitudinal direction and may change during an earthquake. The coefficient values range from 0.3 to 0.8. This study used 0.45 based on previous research [10]. The Mohr–Coulomb failure criterion is used in various geotechnical applications [31]. The material failure criterion depends on the maximum shear stress, and it is assumed that the normal stress regulates the shear stress [24]. Shear stress (τ) is expressed as the cohesion of the material (c), normal stress (σ), and material angle of friction (Φ) and can be defined as follows:

$$\tau = c - \sigma \tan \Phi \tag{5}$$

where τ, c, and Φ change depending on the soil type. This study assumed medium-dense sand as the soil type. According to the longitudinal direction of the applied ground, this study applied the equivalent material property of the medium-dense sand [31].

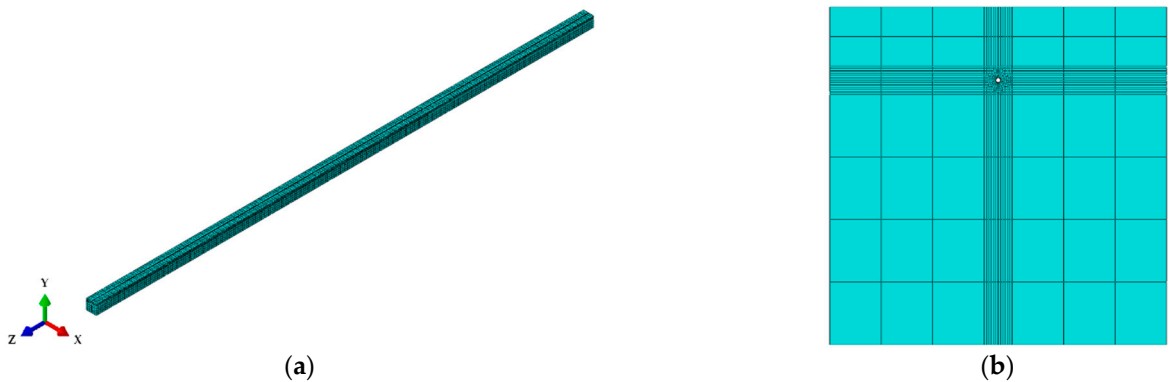

**(a)**

**(b)**

**Figure 12.** FE model of soil: (**a**) isometric view and (**b**) section view.

## 5. Seismic Fragility Analysis

Earthquakes involve several uncertainties in terms of the frequency, PGA, and fault mechanism. Therefore, earthquakes with the same intensity as previous earthquakes cannot occur, i.e., the seismic signatures of earthquakes are distinct. Moreover, the impact of these uncertainties is difficult to identify via conventional deterministic methods. Although probabilistic evaluation requires a larger volume of data than deterministic methods, the recent rapid progress in computational capabilities has rendered a shift from deterministic evaluation to probabilistic evaluation techniques. Therefore, seismic fragility analysis, a probabilistic safety evaluation, has attracted attention, and it is important to assess the seismic performance of various structures [32]. Considering these aspects, this study aims to conduct a seismic fragility analysis that accounts for the uncertainties involved in earthquakes.

### 5.1. Description of Fragility Analysis

The probability of failure ($P_f$) is defined as the conditional probability (*P[A | B])* when the capacity (*C*) of the structure exceeds the seismic demand (*D*) at a specific measure of the seismic intensity (*IM*), as expressed in Equation (6).

$$P_f(IM) = P[C \leq D|IM] \tag{6}$$

In the seismic fragility analysis of the buried pipeline system, PGA, PGV, $PGV^2/PGA$, and peak ground strain were used as the *IM* [33]. PGA is the most commonly used IM in seismic fragility analysis and can be easily acquired using the absolute maximum acceleration of the recorded seismic acceleration history. In several research studies, the seismic fragility analysis of buried pipeline systems was performed by adopting the PGA as the *IM* [34,35]. Considering this parameter's utility, the PGA was selected in this study as the seismic *IM* to perform a fragility analysis. In particular, *C* refers to the set limit state of the structure, including the force, displacement, and stress–strain. Moreover, strain was selected as the limit state of the pipeline system in this study; additional data are mentioned in Section 5.2. Furthermore, *D* represents the structural response quantities that can predict the damage to the structure. By applying Equation (6), the probability of failure at a specific PGA of x is calculated, and the earthquake fragility curve is derived through the function assumed as the cumulative lognormal distribution, as illustrated in Equation (7).

$$P_f(\text{PGA} = \text{x}) = \Phi\left(\frac{\ln(x/\theta_d)}{\beta_d}\right), \tag{7}$$

where $\Phi(\cdot)$ indicates the cumulative lognormal distribution function, $\theta_d$ represents the mean of the lognormal distribution, and $\beta_d$ refers to the standard deviation of the lognormal distribution.

### 5.2. Limit State

Seismic fragility analysis represents the conditional probability that exceeds the set limit state, and the value of the probability of failure may vary depending on the level of the limit state. In other words, the limit state does not necessarily represent the total damage owing to structural collapse and can be classified into several stages as the damage state. Jahangiri and Shakib [12] defined the maximum compressive strain as the limit state of the buried pipeline system. Strain is defined in relation to the diameter and thickness and is classified into four stages. Zhang et al. [36] identified the lack of research for the definition of quantitative damage states of pipelines, citing the study conducted by Shinozuka et al. [37], and applied the limit state of three stages. In this study, the limit state is defined by the relationship between the maximum strain ($\varepsilon_p$) and yielding strain ($\varepsilon_y$), as summarized in Table 4. Particularly, the main factor driving the pipe damage is the ratchet effect, in which plastic strain accumulates owing to low-cycle fatigue; thus, setting the strain as the limit state is considered a reasonable approach.

**Table 4.** Limit state of bellows pipeline system.

| Limit State | Description |
| :---: | :---: |
| None | $\varepsilon_p \leq 0.7\,\varepsilon_y$ |
| Minor | $0.7\,\varepsilon_y < \varepsilon_p \leq \varepsilon_y$ |
| Moderate | $\varepsilon_y < \varepsilon_p \leq 2\,\varepsilon_y$ |
| Major | $2\,\varepsilon_y < \varepsilon_p$ |

### 5.3. Input Motion

Several design codes suggest the number of input motions to be considered in the dynamic analysis of structures, and ASCE 7-05 [38] recommends using at least seven recorded earthquakes [39]. Therefore, in this study, seven earthquakes were selected for analysis considering the uncertainties of earthquakes, as summarized in Table 5. As several incidents of damage to buried pipeline systems have been reported, the 1994 Northridge earthquake and the 1995 Kobe earthquake were included. These two earthquakes were high-frequency earthquakes with dominant frequencies in the domain around 10 Hz, and the influence of earthquakes on the frequency range was considered. The selected input motion was normalized to the PGA of each earthquake, and a nonlinear time history analysis was performed by changing the scale to 0.2 g, 0.6 g, 1.0 g, and 1.5 g. The normalized spectrum of each earthquake was designed as illustrated in Figure 13.

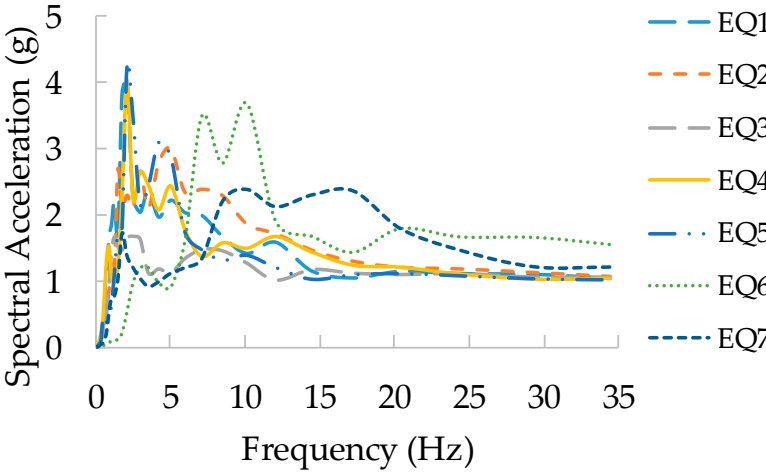

**Figure 13.** Spectral acceleration for input motions.

**Table 5.** Selected input motions.

| No. | Date | Event | Locations | Magnitude (Mw) | PGA (g) |
|---|---|---|---|---|---|
| 1 | 17 January 1994 | Northridge | Beverly Hills, USA | 6.7 | 0.5165 |
| 2 | 17 January 1994 | Northridge | Canyon Country, USA | 6.7 | 0.4820 |
| 3 | 12 November 1999 | Duzce | Bolu, Turkey | 7.1 | 0.8224 |
| 4 | 16 October 1999 | Hector Mine | Hector, USA | 7.1 | 0.3368 |
| 5 | 16 January 1995 | Kobe | Nishi-Akashi, Japan | 6.9 | 0.5093 |
| 6 | 12 September 2016 | Geyongju | Ulsan, Republic of Korea | 5.4 | 0.4425 |
| 7 | 15 November 2017 | Pohang | Pohang, Republic of Korea | 5.5 | 0.2829 |

*5.4. Result*

This study performed the seismic fragility analysis according to the installation condition of the bellows and the comparison between each limit state, as shown in Figure 14. The solid line represents the seismic fragility curve for the piping system without bellows expansion joints, and each dashed line represents the seismic fragility curve for the piping system according to the installation condition of bellows expansion joints.

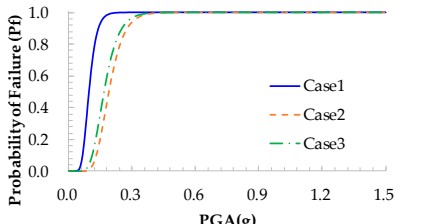 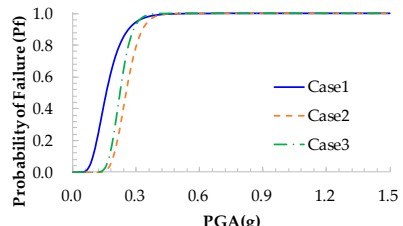 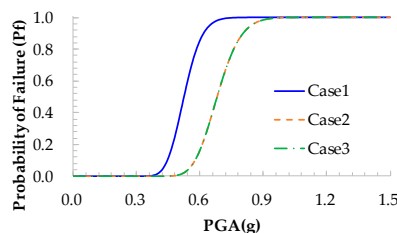

**Figure 14.** Seismic fragility of each case: (**a**) minor; (**b**) moderate; and (**c**) major.

Figure 14a compares the seismic fragility curves for the minor limit state. The PGA at which the probability of failure increased rapidly in each case was 0.04 g, 0.08 g, and 0.06 g, respectively. The median value of each case was 0.095 g, 0.19 g, and 0.167 g, which means that minor damage may occur at a relatively low level of PGA in the piping system without a bellows expansion joint. Figure 14b compares the seismic fragility curve for the moderate limit state. The median value of each case was 0.158 g, 0.251 g, and 0.222 g, respectively, and the moderate limit state was reached at a high PGA when the bellows expansion joint was applied. Figure 14c shows the seismic fragility curves for the major limit state, and the median value of each case was 0.524 g, 0.681 g, and 0.681 g. The median values of Case 2 and Case 3 were the same, and their seismic fragility curves were the same. It seems to be because the response of the piping system was not considered for the uncertainties of various seismic waves due to a relatively small number of input motions. In every case, the probability of failure increased relatively slowly for the major damage state.

Since failure starts at a lower PGA in Case 1 than in Cases 2 and 3 in all limit states, it seems that the bellows expansion joint contributes to the seismic performance improvement of the buried piping system to a certain extent. Furthermore, the median value of Case 3 was similar to or greater than that of Case 2. Through this result, there is a correlation between the number of bellows expansions and the improvement of seismic performance, but the relationship is not linearly proportional.

## 6. Conclusions

The buried piping system is an essential component for transporting energy sources, and failure by external loads may lead to secondary damage, such as explosions or floods caused by leakage of gas, oil, or water. Buried piping systems may be frequently exposed to natural disasters, such as earthquakes and ground subsidence. Since bellows expansion joints allow for some deformation against displacements and rotations, they can respond effectively to large relative displacements caused by earthquakes and ground subsidence.

However, there are very few cases of research on the seismic performance of buried piping system with bellows expansion joints considering the characteristics of earthquakes. In particular, research on seismic fragility analysis using a verified FE model based on cyclic loading test data is non-existent. Furthermore, there are no criteria or recommendations for the installation conditions (such as position and quantity) of bellows expansion joints for the seismic performance improvement of the buried piping system. In this study, therefore, we conducted basic research for the installation guidelines of bellows expansion joints. We developed and validated an FE model of bellows expansion joints based on experimental test data. The following conclusions were derived from the seismic fragility analysis considering the soil-piping system interaction and the bellows installation conditions.

This study performed a cyclic loading test, in which the load increased gradually on the bellows expansion joint and developed and validated an FE model based on the experimental data. The force-displacement observed in the experiment and FE analysis was similar in the overall trend, and the energy dissipation that occurred in each cycle was very similar, with a difference of less than 10%. The experiment ended due to leakage during the unloading process after a displacement of 113.6 mm was applied, and the calculation was also terminated in the FE analysis due to the excessive plastic deformation after a displacement of 113.6 mm was applied. In the experiment, the load decreased gradually while the initial displacement was being applied. However, the phenomenon of decreasing load was not found in the FE analysis. In the case of the convolution, wall-thinning, and unavoidable changes in the material properties occur due to the plastic deformation experienced during the hydroforming process. They may occur differently depending on each convolution. However, the developed FE model applied the same thickness and material properties for all convolutions, considering only the design conditions. Therefore, the variability that could occur in the forming process led to a small difference between the experiment and FE analysis. Furthermore, errors occurred in the FE model because the spring-back phenomenon and the Bauschinger effect were not simulated accurately. In the end, however, the developed model is reliable because the leakage displacement and energy dissipation are similar despite the slight errors.

For the seismic fragility analysis of the buried pipeline system to which bellows are applied, an FE model considering soil-piping system interactions was developed. To analyze the seismic fragility curve according to the application of the bellows expansion joint, three cases were considered: (1) without bellows; (2) with one bellows in the center; (3) with two bellows located 100 m away from both ends. As a result, damage appeared to occur at higher PGA when bellows were applied. Therefore, it is shown that the bellows expansion joint contributes to the improvement of the seismic performance of the buried pipeline system. However, when Case 2 and Case 3 were compared, it was found that Case 3 suffered damage at a lower PGA. It is not that seismic performance increases as the number of bellows expansion joints increases. It seems that the location and distance are more important than the number of bellows applied to the buried pipeline system, and further research is needed to consider parameters such as the location, number, and distance of the bellows. Additionally, depending on the number of convolutions and the thickness of the bellows, there is a difference in performance for axial and angular deformation. Therefore, it is necessary to examine the influence on the geometric properties of the bellows, such as the number and thickness of convolutions, through further research.

**Author Contributions:** Conceptualization, J.-I.R. and B.-S.J.; methodology, H.-Y.S. and B.-S.J.; software, J.-I.R.; validation, B.-S.J. and H.-Y.S.; formal analysis, B.-S.J. and B.-G.J.; investigation, B.-G.J.; resources, B.-G.J. and B.-S.J.; data curation, B.-G.J.; writing—original draft preparation, J.-I.R. and H.-Y.S.; writing—review and editing, J.-I.R. and B.-S.J. All authors have read and agreed to the published version of the manuscript.

**Funding:** This research was funded by the Korea Agency for Infrastructure Technology Advancement(KAIA) grant funded by the Ministry of Land, Infrastructure and Transport of the Korean Government (No. 22CTAP-C164263-02).

**Institutional Review Board Statement:** Not applicable.

**Informed Consent Statement:** Not applicable.

**Data Availability Statement:** The data presented in this study are available on request from the corresponding author.

**Acknowledgments:** The authors gratefully acknowledge support by the research fund of the Korea Agency for Infrastructure Technology Advancement(KAIA) grant funded by the Ministry of Land, Infrastructure and Transport of the Korean Government (No. 22CTAP-C164263-02).

**Conflicts of Interest:** The authors declare no conflict of interest.

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
