# Peer review of "Application of Seismic Fragility of Buried Piping Systems with Bellows Expansion Joints"

_sustainability, doi:10.3390/su142416756_

Round 1
Reviewer 1 Report
This is a very interesting study. An experimental test was performed on bellows expansion joint considering the characteristics of earthquake loading onditions; 3D Finite Element (FE) model using ABAQUS platform was developed and validated based on the experimental results.
The results and discussion contents are too few, and the result data should be properly analyzed in detail. And conclusions and summaries should be more refined.
Author Response
We have modified the manuscript as suggested in the sections.
3.3 Validation of the FE Model
In the experiment, the force in the plus direction showed a tendency of decreasing gradually in the unloading process while the initial displacement was being applied. The maximum force of the first cycle was measured at 1.91 kN, but that of the 15th cycle was measured at 1.10 kN. Although the maximum force decreased by up to approximately 42% at the initial displacement, no decrease in the force was observed in the FE analysis result. During the hydroforming process of bellows, wall-thinning of convolution may occur, and the material may experience plastic deformation[26]. Variations that may occur during the fabrication process can affect the mechanical behavior of bellows. Therefore, the maxi-mum force may decrease as the number of cycles increases. However, in the case of FE analysis, it seems that no decrease in force was observed because the thickness and material properties of bellows were applied based on the design conditions. After the 15th cycle, the displacement increased gradually, reaching 113.6 mm, and the experiment ended be-cause of leakage in the convolution of bellows during the unloading process. In the FE analysis, too, the calculation was terminated because of the excessive plastic deformation during the unloading process after reaching 113.6 mm. A difference in the response occurs between the experiment and analysis because of the uncertainties that may occur in the forming process. Furthermore, it seems that a lower force is generated in the experiment than in the FE analysis because of the Bauschinger effect and the spring back phenome-non. Differences in compressive and tensile stresses occur during the loading and un-loading processes, and this effect is called the Bauschinger effect [27]. Additionally, the energy dissipation occurring in each cycle was calculated, as shown in Figure 10. In most cycles, the calculated energy dissipation in the FE analysis result was larger, with a difference of up to approximately 10%. As the cycle increased, the error decreased gradually, and the energy dissipation of the last cycle was almost the same. In sum, the developed FE model was validated based on the experimental data, and it seems that it can sufficiently simulate the dynamic behavior of bellows.
5.4 Result
This study performed the seismic fragility analysis according to the installation condition of bellows and the comparison between each limit state, as shown in Figure 14. The solid line represents the seismic fragility curve for the piping system without bellows expansion joints, and each dashed line represents the seismic fragility curve for the piping system according to the installation condition of bellows expansion joints.
Figure 14(a) compares the seismic fragility curves for the minor limit state. The PGA at which the probability of failure increased rapidly in each case was 0.04 g, 0.08 g, and 0.06 g, respectively. The median value of each case was 0.095 g, 0.19 g, and 0.167 g, which means that minor damage may occur at a relatively low level of PGA in the piping system without a bellows expansion joint. Figure 14(b) compares the seismic fragility curve for the moderate limit state. The median value of each case was 0.158 g, 0.251 g, and 0.222 g, respectively, and the moderate limit state was reached at a high PGA when the bellows expansion joint was applied. Figure 14(c) shows the seismic fragility curves for the major limit state, and the median value of each case was 0.524 g, 0.681 g, and 0.681 g. The median values of Case 2 and Case 3 were the same, and their seismic fragility curves were the same. It seems to be because the response of the piping system was not considered for the uncertainties of various seismic waves due to a relatively small number of input motions. In every case, the probability of failure increased relatively slowly for the major damage state.
Since failure starts at a lower PGA in Case 1 than in Cases 2 and 3 in all limit states, it seems that the bellows expansion joint contributes to the seismic performance improvement of the buried piping system to a certain extent. Furthermore, the median value of Case 3 was similar to or greater than that of Case 2. Through this result, there is a correlation between the number of bellows expansion and the improvement of seismic performance, but the relationship is not linearly proportional.
- Conclusion
The buried piping system is an essential component for transporting energy sources, and failure by external loads may lead to secondary damage, such as explosion or flood caused by leakage of gas, oil, or water. Buried piping systems may be frequently exposed to natural disasters, such as earthquakes and ground subsidence. Since bellows expansion joints allow for some deformation against displacements and rotations, they can respond effectively to large relative displacements caused by earthquakes and ground subsidence. However, there are very few cases of research on the seismic performance of buried piping system with bellows expansion joints considering the characteristics of earth-quakes. In particular, research on seismic fragility analysis using a verified FE model based on cyclic loading test data is non-existent. Furthermore, there are no criteria or recommendations for the installation conditions (such as position and quantity) of bellows expansion joints for the seismic performance improvement of the buried piping system. In this study, therefore, we conducted basic research for the installation guidelines of bellows expansion joints. We developed and validated an FE model of bellows expansion joints based on experimental test data. The following conclusions were derived from the seismic fragility analysis considering the soil-piping system interaction and the bellows installation conditions.
This study performed a cyclic loading test, in which the load increased gradually on the bel-lows expansion joint and developed and validated an FE model based on the experimental data. The force-displacements observed in the experiment and FE analysis was similar in the overall trend, and the energy dissipation occurred in each cycle was very similar, with a difference of less than 10%. The experiment ended due to leakage during the unloading process after a displacement of 113.6 mm was applied, and the calculation was also terminated in the FE analysis due to the excessive plastic deformation after a displacement of 113.6 mm was applied. In the experiment, the load decreased gradually while the initial displacement was being applied. However, the phenomenon of decreasing load was not found in the FE analysis. In the case of the convolution, wall-thinning and unavoidable changes in the material properties occur due to the plastic deformation experienced during the hydroforming process. They may occur differently depending on each convolution. However, the developed FE model applied the same thickness and material properties for all convolutions, considering only the design conditions. Therefore, the variability that could occur in the forming process led to a small difference between the experiment and FE analysis. Furthermore, errors occurred in the FE model because the spring back phenomenon and the Bauschinger effect were not simulated accurately. In the end, however, the developed model is reliable because the leakage displacement and energy dissipation are similar despite the slight errors.

Reviewer 2 Report
COMMENTS TO THE AUTHOR(S)
This study examined the “Application of Seismic Fragility of Buried Piping System with Bellows Expansion Joint”. Given that earth is a global concern, this paper is timely and could offer new insights on buried pipes system response to earth quake. The manuscript is generally well written and easy to understand. I suggest that the authors revise the manuscript incorporating the following comments and suggestions into an updated version.
1. The authors made several scenarios based on whether bellows were applied or not. Are these the most common technique adopted in the field? Briefly explain it in methodology section.
2. What are the bases for Bellows properties selection? Briefly explain it in methodology section.
3. The authors should assess the goodness of fit between the experimental and FE analysis data.
4. The authors evaluated the seismic performance considering soil–piping system interaction. What type of soil was considered for the analysis? Moreover, how much pipe depth under the soil was considered
5. Did the authors evaluate the seismic performance considering fluid (gas or water flowing in pipe)–piping system interaction.
Author Response
1. The authors made several scenarios based on whether bellows were applied or not. Are these the most common technique adopted in the field? Briefly explain it in methodology section.
Authors Reply:
We thank the reviewer for a careful reading and we revised the manuscript as suggested in the section.
4. Development of Soil-Bellows Piping System Interaction FE Model
In previous studies, performance evaluation was performed on buried piping systems with bellows expansion joints under repeated compressive and shear loads [12], [13]. These studies targeted a short-length buried piping system and evaluated the performance based on the repeated loads of the piping system with and without bellows. However, there are no specific criteria or recommendations for the installation conditions (such as position or quantity) of bellows expansion joints in a long-connected buried piping system. Therefore, we conducted this study as a basic study for providing installation guidelines of bellows expansion joints and analyzed the seismic performance considering the following installation conditions of bellows: (1) without a bellows expansion joint; (2) with a single bellows expansion joint; and (3) with two bellows expansion joints. Figure 11 shows the FE model of the piping system according to the installation condition.
2. What are the bases for Bellows properties selection? Briefly explain it in methodology section.
Authors Reply:
We thank the reviewer for a careful reading and we revised the manuscript as suggested in the section.
3.1 Tensile Coupon Test for STS 316L
The modulus of elasticity was estimated to be 188,000 MPa using the least squares meth-od in the true stress-true strain relationship. A kinematic hardening model was assumed for the material’s hardening. Since the kinematic hardening model is assumed to be a bi-linear type on the ABAQUS Platform [24], the yield and ultimate stress were determined in a such way that the error from the true stress-true strain relation obtained in the experiment would be less than 2%. The yield and ultimate stress were 360 MPa and 1,050 MPa, respectively.
3. The authors should assess the goodness of fit between the experimental and FE analysis data.
Authors Reply:
We thank the reviewer for a careful reading and we revised the manuscript as suggested in the section.
3.3 Validation of the FE Model
In the experiment, the force in the plus direction showed a tendency of decreasing gradually in the unloading process while the initial displacement was being applied. The maximum force of the first cycle was measured at 1.91 kN, but that of the 15th cycle was measured at 1.10 kN. Although the maximum force decreased by up to approximately 42% at the initial displacement, no decrease in the force was observed in the FE analysis result. During the hydroforming process of bellows, wall-thinning of convolution may occur, and the material may experience plastic deformation[26]. Variations that may occur during the fabrication process can affect the mechanical behavior of bellows. Therefore, the maxi-mum force may decrease as the number of cycles increases. However, in the case of FE analysis, it seems that no decrease in force was observed because the thickness and material properties of bellows were applied based on the design conditions. After the 15th cycle, the displacement increased gradually, reaching 113.6 mm, and the experiment ended be-cause of leakage in the convolution of bellows during the unloading process. In the FE analysis, too, the calculation was terminated because of the excessive plastic deformation during the unloading process after reaching 113.6 mm. A difference in the response occurs between the experiment and analysis because of the uncertainties that may occur in the forming process. Furthermore, it seems that a lower force is generated in the experiment than in the FE analysis because of the Bauschinger effect and the spring back phenome-non. Differences in compressive and tensile stresses occur during the loading and un-loading processes, and this effect is called the Bauschinger effect [27]. Additionally, the energy dissipation occurring in each cycle was calculated, as shown in Figure 10. In most cycles, the calculated energy dissipation in the FE analysis result was larger, with a difference of up to approximately 10%. As the cycle increased, the error decreased gradually, and the energy dissipation of the last cycle was almost the same. In sum, the developed FE model was validated based on the experimental data, and it seems that it can sufficiently simulate the dynamic behavior of bellows.
4. The authors evaluated the seismic performance considering soil–piping system interaction. What type of soil was considered for the analysis? Moreover, how much pipe depth under the soil was considered
Authors Reply:
We thank the reviewer for a careful reading and please refer to the section 4 as follows:
The burial depth was assumed to be 1,300 mm, considering the general burial depth of gas pipelines [23]. The coefficient of friction between the soil and the pipeline may vary based on the longitudinal direction and may change during an earthquake. The coefficient values range from 0.3 to 0.8. This study used 0.45 based on previous research [30]. The Mohr–Coulomb failure criterion is used in various geotechnical applications [31]. The material failure criterion depends on the maximum shear stress, and it is assumed that the normal stress regulates the shear stress [24]. Shear stress (τ) is expressed as the cohesion of the material (c), normal stress (σ), and material angle of friction (Φ) and can be defined as follows:
|
τ=c-σtanΦ |
(5) |
where τ, c, and Φ change depending on the soil type. This study assumed medium-dense sand as the soil type. According to the longitudinal direction of the applied ground, this study applied the equivalent material property of the medium-dense sand [31].
5. Did the authors evaluate the seismic performance considering fluid (gas or water flowing in pipe)–piping system interaction.
Authors Reply:
We agree that fluid-piping structure interaction should be considered, but fluid such as gas, oil and water system may include the uncertainties as itself. The problem of bellows piping fragility will be highly sensitive to not only the material and geometrical uncertainties but also soil foundation and ground motion uncertainties. Therefore, as such an issue, the computational cost will be increased. In order to reduce the uncertainties’ issue of the piping system, the fluid condition was replaced as internal pressure into the piping system. We might, however, consider the soil-fluid-piping structure interaction to characterize the seismic performance of the bellows piping system for the future study.

Reviewer 3 Report
In this work, the authors developed a finite element model of bellows expansion joints under seismic conditions by comparison with indoor tests. Then the seismic fragility analysis of the buried pipeline system of bellows expansion joints was performed using simulation software, and the seismic advantages of bellows expansion joints were demonstrated. In general, the workload of this paper is sufficient, and the analysis method is technically sound. However, the structure and content of the article need to be readjusted. It can be published after minor revisions.
Comment 1:
The abstract section needs to be more refined. The background and significance of the study should be briefly summarized. As it stands, it is slightly bloated.
Comment 2:
There are many mistakes in the serial numbers of figures and equations in the manuscript. Please check carefully.
Comment 3:
The article's structure needs to be adjusted. The manuscript devotes much space to the details of the experiment's methodology and model building, but the results' analysis needs to be more prominent. Key steps should be selected for a detailed presentation and the rest as auxiliary. Academic research should be oriented toward scientific issues rather than research methods.
Comment 4:
What is the point of the article? Is it to present the rationality of the model developed or to explore the performance of bellows expansion joints in seismic environments?
Comment 5:
The article's conclusion needs to be organized and should briefly describe the work of the paper, highlighting the results obtained from experiments or simulations.
Author Response
1. The abstract section needs to be more refined. The background and significance of the study should be briefly summarized. As it stands, it is slightly bloated.
Authors Reply:
We thank the reviewer for a careful reading and we revised the abstract.
Abstract
Bellows expansion joint is known to have large displacement capacity and can be thus potentially used to improve the seismic performance of the buried piping system. However, there are no guidelines on the installation of bellows expansion joints for seismic performance improvement of buried piping systems. Furthermore, there are very few studies on the seismic performance of buried piping systems with bellows expansion joints. In this study, therefore, we per-formed seismic fragility analysis according to the installation conditions to obtain basic data for installation guidelines of bellows expansion joints.
2. There are many mistakes in the serial numbers of figures and equations in the manuscript. Please check carefully.
Authors Reply:
We thank the reviewer for a careful reading and for catching this typo. The numbers didn’t correspond to each other due to incorrect numbers being used. We have now fixed this typo and updated the manuscript.
3. The article's structure needs to be adjusted. The manuscript devotes much space to the details of the experiment's methodology and model building, but the results' analysis needs to be more prominent. Key steps should be selected for a detailed presentation and the rest as auxiliary. Academic research should be oriented toward scientific issues rather than research methods.
Authors Reply:
We believe that this comment is embedded in the current philosophy of piping analysis and validate the FE model of the piping system. Then, we would like to address a few key items that form the basis of this study.
5.4 Result
This study performed the seismic fragility analysis according to the installation condition of bellows and the comparison between each limit state, as shown in Figure 14. The solid line represents the seismic fragility curve for the piping system without bellows expansion joints, and each dashed line represents the seismic fragility curve for the piping system according to the installation condition of bellows expansion joints.
Figure 14(a) compares the seismic fragility curves for the minor limit state. The PGA at which the probability of failure increased rapidly in each case was 0.04 g, 0.08 g, and 0.06 g, respectively. The median value of each case was 0.095 g, 0.19 g, and 0.167 g, which means that minor damage may occur at a relatively low level of PGA in the piping system without a bellows expansion joint. Figure 14(b) compares the seismic fragility curve for the moderate limit state. The median value of each case was 0.158 g, 0.251 g, and 0.222 g, respectively, and the moderate limit state was reached at a high PGA when the bellows expansion joint was applied. Figure 14(c) shows the seismic fragility curves for the major limit state, and the median value of each case was 0.524 g, 0.681 g, and 0.681 g. The median values of Case 2 and Case 3 were the same, and their seismic fragility curves were the same. It seems to be because the response of the piping system was not considered for the uncertainties of various seismic waves due to a relatively small number of input motions. In every case, the probability of failure increased relatively slowly for the major damage state.
Since failure starts at a lower PGA in Case 1 than in Cases 2 and 3 in all limit states, it seems that the bellows expansion joint contributes to the seismic performance improvement of the buried piping system to a certain extent. Furthermore, the median value of Case 3 was similar to or greater than that of Case 2. Through this result, there is a correlation between the number of bellows expansion and the improvement of seismic performance, but the relationship is not linearly proportional.
4. What is the point of the article? Is it to present the rationality of the model developed or to explore the performance of bellows expansion joints in seismic environments?
Authors Reply:
We respectfully agree with the basis of this comment. In general, the experimental test in terms of soil-piping structure interaction is costly and has space-limited to evaluate the seismic performance and also in the case of seismic fragility, big data such as ground motion uncertainties and material uncertainties must be required. Therefore, the analysis using FE model reconciled with the experimental results is more economical to evaluate the performance of the piping system rather than the experimental tests. In particular, the damage of piping system is mostly caused by repetitive large relative displacement under seismic ground motions. So, the FE model based on the results of cyclic loading tests as an experimental test should be necessary. However, previous studies were mainly limited to bellows expansion joints under the condition of monotonic loads and there exists only few the study related to the seismic performance of bellows expansion joints without the development or validation of the bellows piping system using the experimental test. In addition, there was no studies on seismic fragility analysis corresponding to the uncertainty of seismic ground motion in a buried piping system with bellows expansion joints. Also, installation standards or regulatory guidelines for applying bellows expansion joints to buried piping system were rarely mentioned. Consequently, the primary objective of this study was to distribute the basic data to propose the guideline associated with the installation of the bellows expansion joints to mitigate the earthquake damage of the buried piping system. In order to conduct the seismic fragility of piping system with bellows expansion joints under strong ground motions, reconciliation of the analytical and experimental results was addressed in this study. Then, the seismic fragilities considering soil-piping structure interaction was generated. Finally, the seismic performance of the piping system with and without bellows expansion joints and by the location of the bellows was compared in this study.
5. The article's conclusion needs to be organized and should briefly describe the work of the paper, highlighting the results obtained from experiments or simulations.
Authors Reply:
We thank the reviewer for a careful reading and we revised the manuscript as suggested in the section.
6. Conclusion
The buried piping system is an essential component for transporting energy sources, and failure by external loads may lead to secondary damage, such as explosion or flood caused by leakage of gas, oil, or water. Buried piping systems may be frequently exposed to natural disasters, such as earthquakes and ground subsidence. Since bellows expansion joints allow for some deformation against displacements and rotations, they can respond effectively to large relative displacements caused by earthquakes and ground subsidence. However, there are very few cases of research on the seismic performance of buried piping system with bellows expansion joints considering the characteristics of earth-quakes. In particular, research on seismic fragility analysis using a verified FE model based on cyclic loading test data is non-existent. Furthermore, there are no criteria or recommendations for the installation conditions (such as position and quantity) of bellows expansion joints for the seismic performance improvement of the buried piping system. In this study, therefore, we conducted basic research for the installation guidelines of bellows expansion joints. We developed and validated an FE model of bellows expansion joints based on experimental test data. The following conclusions were derived from the seismic fragility analysis considering the soil-piping system interaction and the bellows installation conditions.
This study performed a cyclic loading test, in which the load increased gradually on the bellows expansion joint and developed and validated an FE model based on the experimental data. The force-displacements observed in the experiment and FE analysis was similar in the overall trend, and the energy dissipation occurred in each cycle was very similar, with a difference of less than 10%. The experiment ended due to leakage during the unloading process after a displacement of 113.6 mm was applied, and the calculation was also terminated in the FE analysis due to the excessive plastic deformation after a displacement of 113.6 mm was applied. In the experiment, the load decreased gradually while the initial displacement was being applied. However, the phenomenon of decreasing load was not found in the FE analysis. In the case of the convolution, wall-thinning and unavoidable changes in the material properties occur due to the plastic deformation experienced during the hydroforming process. They may occur differently depending on each convolution. However, the developed FE model applied the same thickness and material properties for all convolutions, considering only the design conditions. Therefore, the variability that could occur in the forming process led to a small difference between the experiment and FE analysis. Furthermore, errors occurred in the FE model because the spring back phenomenon and the Bauschinger effect were not simulated accurately. In the end, however, the developed model is reliable because the leakage displacement and energy dissipation are similar despite the slight errors.

Round 2
Reviewer 3 Report
Please polish the language of the paper again!